# Epitope Mapping of Monoclonal Antibodies to Calreticulin Reveals That Charged Amino Acids Are Essential for Antibody Binding

**DOI:** 10.3390/antib10030031

**Published:** 2021-08-04

**Authors:** Ann Christina Bergmann, Cecilie Kyllesbech, Rimantas Slibinskas, Evaldas Ciplys, Peter Højrup, Nicole Hartwig Trier, Gunnar Houen

**Affiliations:** 1Department of Autoimmunology, Statens Serum Institut, 2300 Copenhagen, Denmark; ann_bergmann@hotmail.com; 2Department of Neurology, Rigshospitalet Glostrup, 2600 Glostrup, Denmark; cekr16@student.sdu.dk; 3Institute of Biotechnology, University of Vilnius, 01513 Vilnius, Lithuania; rimantas.slibinskas@bti.vu.lt (R.S.); evaldas.ciplys@bti.vu.lt (E.C.); 4Department of Biochemistry and Molecular Biology, University of Southern Denmark, 5230 Odense, Denmark; php@bmb.sdu.dk

**Keywords:** antibody, calreticulin, epitope, enzyme-linked immunosorbent assay

## Abstract

Calreticulin is a chaperone protein, which is associated with myeloproliferative diseases. In this study, we used resin-bound peptides to characterize two monoclonal antibodies (mAbs) directed to calreticulin, mAb FMC 75 and mAb 16, which both have significantly contributed to understanding the biological function of calreticulin. The antigenicity of the resin-bound peptides was determined by modified enzyme-linked immunosorbent assay. Specific binding was determined to an 8-mer epitope located in the N-terminal (amino acids 34–41) and to a 12-mer peptide located in the C-terminal (amino acids 362–373). Using truncated peptides, the epitopes were identified as TSRWIESK and DEEQRLKEEED for mAb FMC 75 and mAb 16, respectively, where, especially the charged amino acids, were found to have a central role for a stable binding. Further studies indicated that the epitope of mAb FMC 75 is assessable in the oligomeric structure of calreticulin, making this epitope a potential therapeutic target.

## 1. Introduction

Calreticulin (CRT) is a molecular chaperone protein mainly located in the endoplasmic reticulum (ER), where it also functions as a Ca^2+^ storage protein [1,2]. CRT has been assigned multiple other biological and immunological functions in addition to the above mentioned, including a role as a surface opsonin (“eat me signal”) in connection with surface translocation during so-called “immunogenic cell death” [1,3,4,5,6,7,8]. Moreover, characteristic mutations in the C-terminus of CRT, mainly insertions and deletions (INDELS), cause a frameshift (fs) change to a common multibasic sequence (CRTfs), which has been found to be involved in several myeloproliferative neoplasms (MPNs) [9,10,11,12,13].

The structure of CRT is unusual. The N-terminal 200 amino acids (aas) form the major part of a β sandwich, from which a proline-rich hairpin of approximately 100 aas protrudes. This domain continues into a C-terminal part of another 100 aas, which contributes two strands to the β sandwich and an α-helix, which together with the N-terminal part constitutes a globular core. The rest of the C-terminus is strongly acidic and can bind multiple Ca^2+^, thus stabilizing the molecule, and it ends in a KDEL ER retention signal [14,15,16,17,18]. Both wild-type (wt) CRT and CRTfs can form oligomers, a property that is important for the properties of the molecules [19,20,21,22,23,24].

Antibodies (Abs) in general are crucial reagents in the study of the biological and immunological functions of proteins and other macromolecules, and monoclonal Abs (mAbs) and peptide Abs have revolutionized molecular and cellular biology [25,26,27,28]. The importance of Abs as reagents applies in particular to CRT and CRTfs, due to their unusual structures and many diverse functions. Several Abs have been described, and some are commercially available, but none have been adequately characterized in relation to interaction with specific parts of the chaperone, although some knowledge of their epitopes is available [29,30,31]. Here, we report the epitope mapping of two important mAbs to CRT, which have contributed to understanding the biological function of CRT [32,33,34,35,36,37,38].

Although several studies have been described using the two selected mAbs, the aas crucial for antigen (Ag) binding still remain to be characterized. Moreover, the study of theseAbs may contribute to determining the structure of the CRT protein, as only a part of CRT has been crystalized [16,18].

## 2. Materials and Methods

### 2.1. Materials

All synthetic peptides were purchased from Schäfer-N, Copenhagen, Denmark. CRT mAb FMC 75 (Cat.no. ADI-SPA-601) and mAb 16 (Cat.no. 612137) were purchased from Enzo Life Sciences and BD Transduction laboratories, respectively. MAb FMC 75 is a mouse IgG1 directed to full-length recombinant CRT, whereas mAb 16 is a mouse IgG1 directed to the C-terminal of CRT (aas 270–390). Both Abs were generated in mice using traditional immunization techniques for Ab production, where the mice were immunized with recombinant full-length CRT or a protein fragment. Tris-Tween-NaCl (TTN) (0.3 M NaCl, 20 mM Tris, 0.01% Tween 20, pH 7.5), carbonate buffer (0.05 M sodium carbonate, pH 9.6), alkaline phosphatase (AP) substrate buffer (1 M ethanolamine, 0.5 mM MgCl_2_, pH 9.8), native sample buffer (Tris-HCL 0.5M pH 8.6, glycerol 99%, bromphenylblåt 0.1% *w/v*), mouse IgG1 mAbs directed to β-galactosidase and to BAM were from Statens Serum Institut (Copenhagen, Denmark). *Para*-nitrophenyl phosphate (*p*-NPP) substrate tablets, BCIP/NTB tablets, and goat anti-mouse IgG-AP were purchased from Sigma Aldrich (Steinheim, Germany). Recombinant CRT was produced using standard protocols and purified as previously described by Čiplys and colleagues [39]. Precision Plus Protein Standard was purchased from BIO-RAD. Native PAGE running buffer (pH 7.5), 4–20% Tris-glycine gels and, PVDF membranes were from Thermo–Fisher (Waltman, MA, USA).

### 2.2. Design of Synthetic Peptides

The human CRT sequence (Prot id: P27797), comprising 400 aas (without the N-terminal signal peptide), was used to generate overlapping peptides. The peptides were 20 aas long, each with an overlap of 10 aas (Table A1).

To determine essential terminal aas, N- and C-terminally truncated peptides were applied (Table 1), whereas alanine and functionally substituted peptides were used to determine essential aas crucial for Ab binding (Table 1). All peptides were synthesized and tested on the resin, which allows rapid epitope identification [40,41,42]. The peptides were synthesized on a TentaGel resin using standard solid-phase peptide synthesis.

### 2.3. Screening of Antibody Reactivity to Wild Type Calreticulin by Enzyme-Linked Immunosorbent Assay

CRT (1 µg/mL) diluted in carbonate buffer was added to Polysorp microtiter plates (Thermo–Fisher, Waltman, MA, USA) and incubated overnight (ON) at room temperature (RT). Wells were rinsed with TTN (200 µL/well) and blocked in TTN buffer for 1 h (h) at RT on a shaking table. CRT mAb (1 µg/mL), diluted in TTN buffer, were added to the microtiter wells and incubated for 1 h at RT on a shaking table. Hereafter, the wells were rinsed with TTN (200 µL) for 1 min, and this was repeated 3 times. Next, AP-conjugated goat anti-mouse IgG (1 µg/mL) was added and incubated for 1 h at RT followed by washing as above. Bound Abs were quantified using 1 mg/mL *p*-NPP in AP substrate buffer, 100 µL per well. The absorbance was measured at 405 nm, with background subtraction at 650 nm on a ThermoMax Microtiter Plate Reader from Molecular Devices (San Jose, CA, USA). All samples were tested in duplicates and corrected for background noise.

### 2.4. Screening of Resin-Bound Peptides by Modified Enzyme-Linked Immunosorbent Assay

Resin-bound peptides (100 µg/mL) were added to a 96-well multiscreen filter plate (Millipore, Copenhagen, Denmark) and blocked in TTN for 15 min. CRT mAbs were diluted to a final concentration of 1 µg/mL in TTN buffer, added to the microtiter wells, and incubated for 1 h at RT. Hereafter the wells were rinsed with TTN (200 µL) for 1 min, and this was repeated three times. Next, AP-conjugated goat anti-mouse IgG diluted to 1 µg/mL was added and incubated for 1 h at RT followed by washing as above. Bound Abs were quantified using 1 mg/mL *p*-NPP in AP substrate buffer, 100 µL pr well. After a satisfying color reaction, the buffer was transferred to a MaxiSorp Microtiter plate, and the absorbance was measured at 405 nm, with background subtraction at 650 nm on a ThermoMax Microtiter Plate Reader. All samples were tested in duplicates and corrected for background noise.

### 2.5. PAGE and Western Blotting

For Coomassie staining, recombinant CRT and CRT purified from human placenta were diluted in native sample buffer (1:2) and incubated for 10 min at 57 °C. Ten µL samples were loaded into wells of a 4–20% Tris-glycine gel and run for approximately 75 min at 150 V using native PAGE running buffer. Gels were stained ON at 4 °C with Coomassie Brilliant Blue and washed with Milli-Q water 5 times until bands were visualized.

For Western blotting, pretreated (57 °C for 10 min) recombinant CRT (1 mg/mL) (with and without 42 °C for three days) or CRT (1 mg/mL) purified from human placenta were loaded to a 4–20% Tris-glycine gel and run for approximately 75 min at 150 V using native PAGE running buffer, as described above.

After electrophoresis, gels were blotted onto a PVDF membrane using an iBlotter (Thermo–Fisher, Waltman, MA, USA). Next, the membranes were blocked ON in TTN buffer and mounted in a mini blotting device. Mab FMC 75 was diluted at 1:10,000 in TTN, added to each well, and incubated for 1 h at RT. Next, the membrane was washed for 3 × 5 min in TTN, whereafter AP-conjugated goat anti-mouse IgG, diluted at 1:2000, was added and incubated for 1 h on the membrane followed by washing 3 × 5 min in TTN. Finally, AP substrate (BCIP 0.5 mg/mL, NTB 0.3 mg/mL) was added and incubated for approximately 10 min. The reaction was stopped by washing the membrane in Milli-Q water, followed by drying it on filter paper.

## 3. Results

### 3.1. Reactivity of Monoclonal Antibodies to Wild Type Calreticulin

To analyze the specific reactivity of two selected mAbs to CRT, Ab reactivity to full-length CRT was determined by ELISA. A mAb generated in the same host but with irrelevant specificity was used as a negative control.

As seen in Figure 1, the two mAbs showed specific reactivity to the full-length recombinant wt CRT, as neither the negative control Ab nor the secondary Ab reacted with the CRT protein.

### 3.2. Reactivity of Calreticulin Antibodies to Overlapping Synthetic Peptides

To identify the antigenic regions of the two mAbs, Ab reactivity to overlapping peptides covering the full-length CRT protein was determined by modified ELISA using resin-bound peptides.

As seen in Figure 2, mAb FMC 75 recognized peptide 2 in the N-terminus, whereas mAb 16 recognized a peptide in the C-terminus, corresponding to peptide 35. Peptide 2 corresponds to aas 28–47, whereas peptide 35 corresponds to aas 358–377 in the native CRT sequence. To verify that the interaction to peptide 2 and 35 was specific, a mAb of irrelevant specificity was used as a negative control. As seen, no reactivity to the overlapping CRT peptides was found, indicating that the interactions of mAb FMC 75 and mAb 16 to peptides 2 and 35, respectively, were specific.

Based on the current findings, peptide 2 (LDGDGWTSRWIESKHKSDFG) and 35 (KDKQDEEQRLKEEEEDKKRK) were selected for epitope identification.

### 3.3. Identification of Minimal Sequences for Antibody Binding

To determine the minimal peptide length required for Ab binding, resin-bound peptides truncated from the N- and C-terminal ends were screened for Ab reactivity by modified ELISA. Peptides GDGWTSRWIESKHKSD (2) and KQDEEQRLKEEEEDKK (35) were used as templates for the generation of truncated peptides and functioned as the positive controls.

Figure 3a,b illustrates the reactivity of mAb FMC 75 to N- and C-terminally truncated peptides. As seen, the N-terminal aa Thr was crucial for Ab binding as no Ab reactivity was found to the SRWIESKHKSD peptide compared to the TSRWIESKHKSD peptide. Similarly, the C-terminal aa Lys was found to be crucial for Ab binding, as no reactivity was found to the GDGWTSRWIES peptide. Based on the current findings, the sequence TSRWIESK was determined to be the epitope of mAb FMC 75.

Figure 3c,d illustrate the reactivity of mAb 16 to N- and C-terminally truncated peptides. As seen, the terminal Asp in positions 3 and 14 of the 16-mer template peptide were crucial, respectively, as peptides that did not contain one of these aas did not react with the mAb 16. Based on the current findings, the sequence DEEQRLKEEED was identified as the epitope of mAb 16.

### 3.4. Determination of Antigenic Amino Acids Essential for Antibody Reactivity

The preliminary analysis identified the sequences TSRWIESK and DEEQRLKEEEED as the epitopes of mAb FMC 75 and mAb 16, respectively. To identify aas essential for binding in the identified epitopes, alanine scanning was conducted, where each aa in the identified epitopes of mAb FMC 75 and mAb 16 was substituted with alanine one at a time.

Figure 4a illustrates the reactivity of mAb FMC 75 to alanine-substituted peptides, using the peptide TSRWIESK as a template. As seen, the aas Thr^1^, Ser^2^, Arg^3^, Trp^4^, and Lys^8^ identified relative to the template peptide were essential for Ab binding. The last three residues either have the same or higher intensity compared to the control peptide.

Figure 4b illustrates the reactivity of mAb 16 to alanine-substituted peptides analyzed by modified ELISA. As seen, the majority of the alanine-substituted peptides did not react with the mAb 16; only the substitution of Glu^2^ and Leu^6^ was tolerated. Furthermore, the Ab binding was reduced by approximately 40% when Glu^10^ was substituted with Ala.

Next, to determine whether the dependency of the individual aa residue relates to the functional group in the side chain, a functionality scan was conducted, where the specific contributions of the crucial aa side chains were determined by replacing the aas with an aa of similar functionality.

Figure 4c illustrates the reactivity of mAb FMC 75 to functionality-substituted peptides. As seen, Ab reactivity was reduced when Ser^2^ was substituted with Thr and when Trp^4^ was substituted with Phe. Furthermore, Ab reactivity to the Lys^8^ was reduced by approximately 40% when substituted to Arg when compared with the positive control.

Figure 4d illustrates the reactivity of mAb 16 to functionality-substituted peptides. As seen, Ab reactivity was significantly reduced when replacing the majority of the aas compared to the control peptides. Only the substitution of Glu^2^ with Asp did not reduce Ab binding, whereas the binding was reduced by 60–100% for the remaining residues relative to the control.

Collectively, the characterized Abs have a high dependency on the majority of the aas represented in the epitope.

### 3.5. Epitope Presentation in the Native Calreticulin Structure

Finally, the native structures of the identified epitopes were determined using a CRT crystal structure covering aas 16–367, originally identified by Chouquet et al. [16].

As seen in Figure 5a, the native epitope of mAb FMC 75 (yellow) was found in a flexible β-strand structure. The essential aas Thr, Ser, Arg, and Lys in the mAb FMC 75 epitope protruded in the crystal structure, whereas the Trp residue pointed into the core structure of CRT (Figure 5b).

Unfortunately, the complete structure of CRT has not been crystallized, and hence only the N-terminal region of the mAb 16 epitope (aa 263–373) could be identified. However, the first five aas of the C-terminal epitope were identified to be located in an α-helix structure. Moreover, as presented in Figure 5c, α-helical wheel representation of mAb 16 indicates that the complete epitope is found in an α-helix structure, where residue 2, 6, and 10 were seen to be located on the same side of the potential α-helix.

### 3.6. Reactivity of mAb FMC 75 to Oligomerized Calreticulin

As presented, the epitope of FMC 75 was located in the N-terminal; in contrast, the epitope of mAb 16 was located in the C-terminal end, where potential fs mutations occasionally may occur. To determine whether the epitope of mAb FMC 75 has therapeutic potential, a Coomassie staining and a Western blot were conducted to examine whether the epitope of mAb FMC 75 is assessable upon oligomerization, as the oligomeric structure of CRT is essential for some of the protein’s functions.

As presented in Figure 6a, CRT formed oligomers upon heat treatment, which were recognized by the mAb FMC 75 (Figure 6b), suggesting that the epitope of mAb FMC 75 could be a potential therapeutic target, as it is assessable in the oligomeric structure.

## 4. Discussion

This study describes the specific reactivity of mAb FMC 75 and mAb 16, which both have contributed to elucidate the biological function of CRT, among others, through detection of CRT exposure in the course of immunogenic cell death and examination of the biophysical interaction with other target molecules [33,34,38]. Moreover, FMC 75 has contributed to studies describing tumor-associated CRT variants [36].

In this study, both Abs recognized the wt CRT in ELISA, and as shown in Western blotting, the mAb FMC 75 recognized monomers as well as oligomers. These findings indicate that the epitope of the mAb FMC 75 is accessible upon oligomerization, which is critical for some CRT functions [19], and that the epitope of FMC 75 could be a potential therapeutic target, e.g., in myeloproliferative diseases. In contrast, the epitope of mAb 16 was identified to be located in the C-terminal end, which is prone to mutations causing fs in the C-terminus. Hence, the value of mAb 16 as a potential therapeutic is more limited.

The two peptide-specific CRT Abs recognized epitopes in the N- or C-terminal end of wt CRT, thus, the CRT mAbs reacted to the synthetic peptides as well as the native Ags (Figure 1 and Figure 2), which is a crucial factor defining a well-functioning Ab [25,26,27]. Moreover, these findings indicate that the epitopes of the CRT mAbs are located in surface-accessible and flexible regions, which is typical for many Abs recognizing linear epitopes [40,42,43] and which is confirmed by comparison with published CRT structures [16,17].

Using overlapping peptides covering the complete CRT sequence, it was possible to identify the antigenic regions of mAb FMC 75 and mAb 16 (Figure 2), an approach which often has been applied with success for Ab characterization [42,43,44,45], as screening for Ab reactivity directly on the resin allows rapid identification [40,41,42].

Using N- and C-terminally truncated peptides, the essential terminal aas for both epitopes were determined, corresponding to eight residues in the N-terminus for mAb FMC 75 (34–41, TSRWIESK) and 12 residues in the C-terminus for mAb 16, corresponding to aa 362–373 (DEEQRLKEEEED) (Figure 3). For both epitopes, the charged aas were essential for binding. The two positively charged aas in the epitope of mAb FMC 75 (Arg and Lys) could not be substituted with Ala without reducing Ab binding; however, it was possible to substitute both with aas containing a positive charge and still obtain a significant level of reactivity, indicating that the presence of a positively charged aa rather than the specific aa was essential for Ab binding. Similar to mAb 16, charged aas appeared to prevail in the interaction, as most of the aas found in the epitope of mAb 16 were positively or negatively charged aas (Figure 4). Alanine substitution was only tolerated in positions 2 and 6 (Glu and Leu). As the peptide is believed to be in α -helical conformation, this fits with residues 2, 6, and 10 being located on the same side of the α-helix as shown by an α-helical wheel. However, when substituting Glu2 and Leu6 with aas of similar functionality (Gln and Ile, respectively), Ab reactivity was still obtained. Collectively, these findings indicated that Glu and Leu in positions 2 and 6, respectively, are not crucial for Ab binding and presumably are located in the part of the α-helix contacting the rest of the molecule (i.e., away from the solvent). As presented in Figure 5c, the N-terminal region of the mAb 16 epitope was located in an α-helix structure. Although the complete epitope could not be determined from the crystal structure, recent findings have proposed that the missing region of the epitope is found in an α-helix structure as well [17,46,47]. Collectively, these findings contribute to increased knowledge of the crystal structure of the C-terminal end of CRT, as only the first 365 aas of CRT have been crystallized.

It remains to be determined whether all aas in the epitope are directly involved in the Ab- Ag interaction, as the aas side chains protrude in different directions around the α-helix structure. The most reasonable explanation is that some of the charged aas make direct contact with the Ab, whereas the remaining aas function to stabilize the helix structure.

Whereas most of the aa residues of epitope 2 were essential for the Ab-Ag interaction, the aas of epitope 1 (34–41, TSRWIESK) were more tolerant to substitutions. Besides the dependency of the positively charged aas already mentioned, the epitope of mAb FMC 75 was found to depend on other factors for binding as well. Thus, Thr in position 1 appeared to contribute a hydroxy group for the formation of a hydrogen bond, as Thr could not be substituted with Ala but could be substituted with Ser. Similarly, Ser in position 2 could not be substituted with either Thr or Ala, indicating that the side-chain of Ser in position 2 is essential for Ab binding. Trp in position 4 could not be substituted with either Ala or Phe, indicating that this aa is essential for binding as well. However, when comparing the results to the crystal structure, it was found that the Trp residue points into the core structure of CRT, suggesting that Trp does not make direct contact with mAb FMC 75 but functions to stabilize the epitope. The aas in positions 5–7 could be substituted with Ala or aas with similar functionality, indicating that these aas contribute with backbone conformation and stability or hydrogen bonds to the Ab-epitope interface. Collectively, these findings indicate that charged aas and aas contributing with hydrogen bonds are essential for Ab binding.

The versatile nature of the specific contribution of the individual aas in the Ab-epitope interface is in accordance with the literature [40,41,42,43,44,45]. Several examples of characterized Ab-Ag interactions illustrate that the contribution of the epitope is unique for the specific complex [40,41,42,43,44,45]. Thus, some interactions appear to be backbone-dependent, some appear to depend on charged and hydrophilic aas, as described in this study, and some depend on hydrophobic aas for a stable interaction [40,41,42,43,44,45,48,49]. This clearly illustrates the versatile nature of Abs and Ab-Ag interactions, which is in accordance with the findings described in this study.

## Figures and Tables

**Figure 1 antibodies-10-00031-f001:**
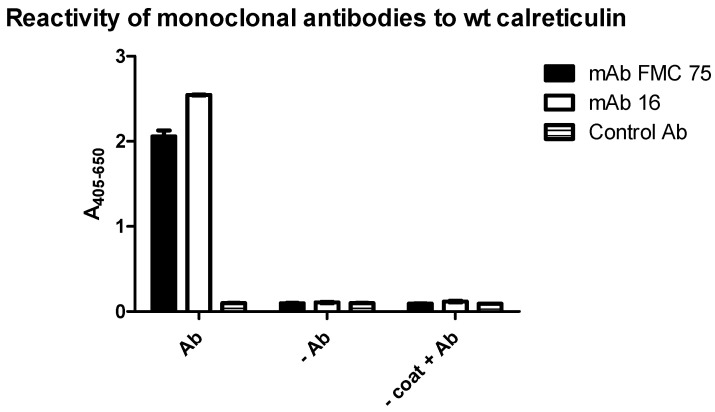
Antibody reactivity to wild-type calreticulin analyzed by enzyme-linked immunosorbent assay. Mouse anti-BAM was used as a negative control. −Ab functions as a control for the secondary antibody goat-anti mouse IgG. − Coat + Ab was added to verify specific reactivity.

**Figure 2 antibodies-10-00031-f002:**
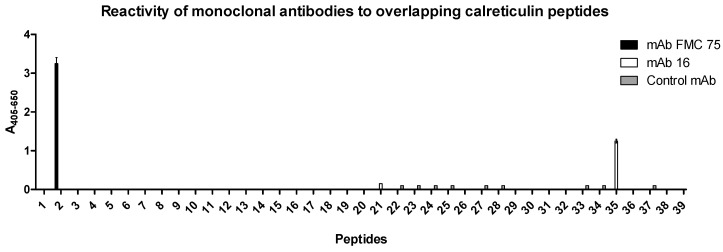
Antibody reactivity to overlapping calreticulin peptides analyzed by modified enzyme-linked immunosorbent assay using resin-bound peptides. Mouse anti-β-galactosidase was used as a negative control.

**Figure 3 antibodies-10-00031-f003:**
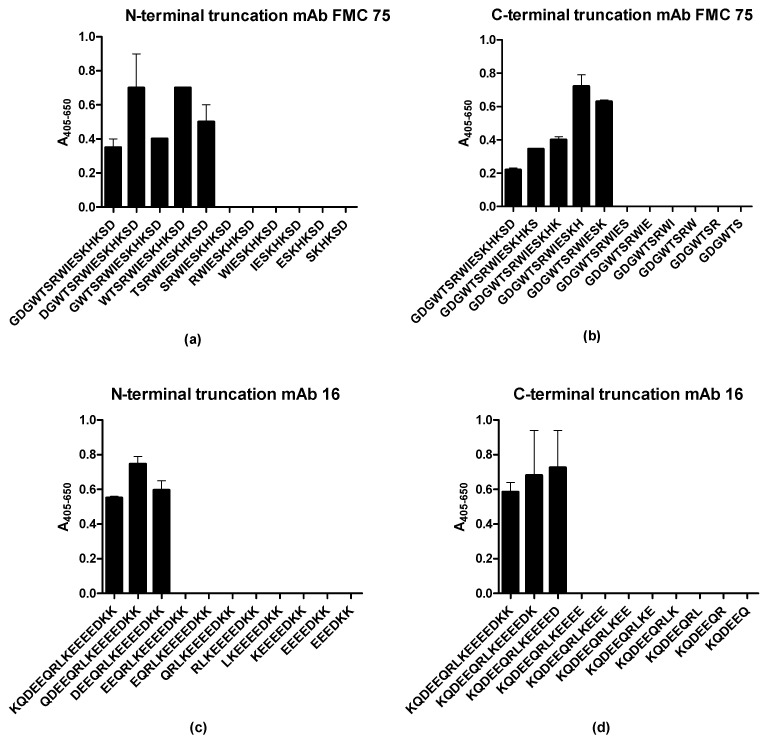
Antibody reactivity to N- and C-terminally truncated peptides. (**a**) Reactivity of mAb FMC 75 to N-terminal truncated peptides. (**b**) Reactivity of mAb FMC 75 to C-terminal truncated peptides. (**c**) Reactivity of mAb 16 to N-terminal truncated peptides. (**d**) Reactivity of mAb 16 to C-terminal truncated peptides.

**Figure 4 antibodies-10-00031-f004:**
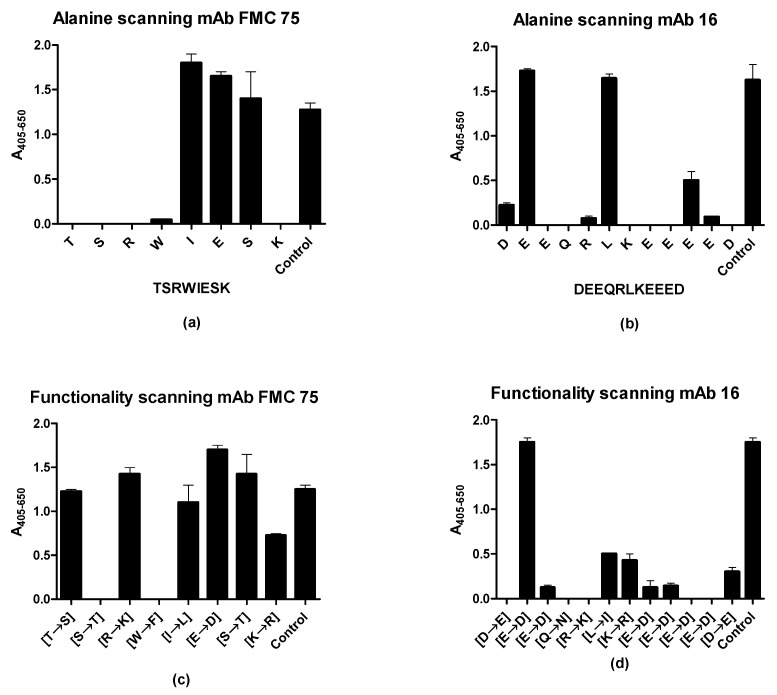
Antibody reactivity to alanine- and functionality-substituted peptides. (**a**) Reactivity of mAb FMC 75 to alanine-substituted peptides. (**b**) Reactivity of mAb 16 to alanine-substituted peptides. (**c**) Reactivity of mAb FMC 75 to functionality-substituted peptides. (**d**) Reactivity of mAb 16 to functionality-substituted peptides.

**Figure 5 antibodies-10-00031-f005:**
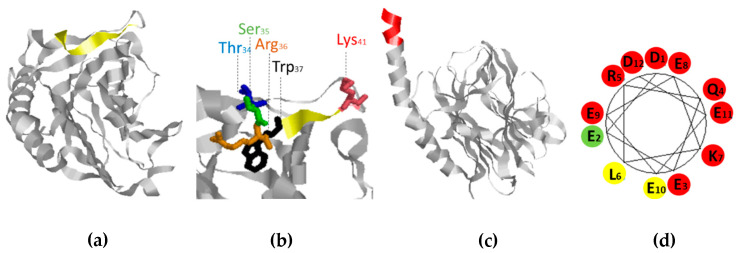
Visualization of the identified epitopes in the native structure of calreticulin. (**a**) Visualization of the mAb FMC 75 epitope. (**b**) Visualization of the essential amino acids in the mAb FMC 75 epitope (amino acids 34–). (**c**) Visualization of the N-terminal region (363–367) of the mAb 16 epitope (362–373). (**d**) α-helical wheel representation of mAb 16. “Red” represents aas which are critical for antibody binding. ”Yellow” represents aas semi-critical aas. “Green” represents non-critical aas residues.

**Figure 6 antibodies-10-00031-f006:**
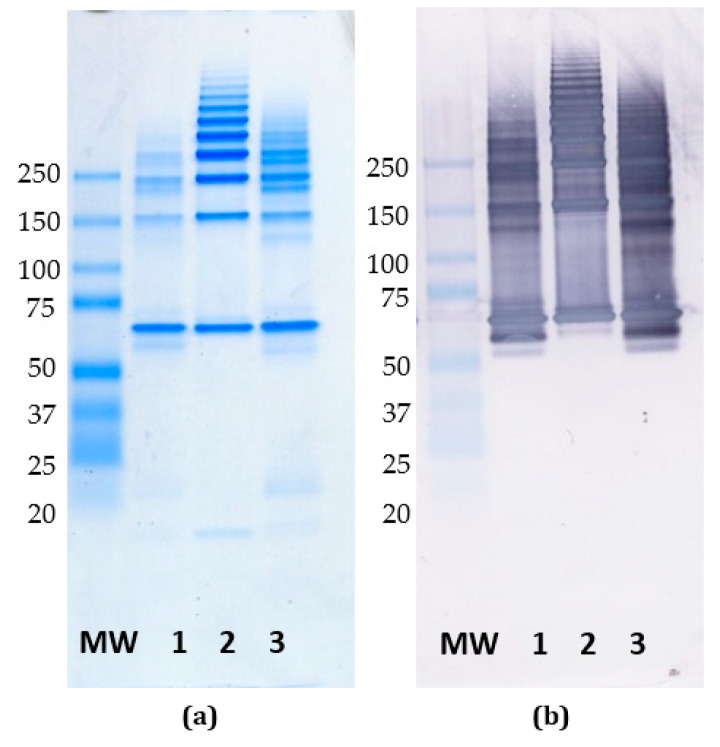
Native PAGE and Western blotting of wild-type calreticulin. (**a**) Coomassie staining of oligomerized calreticulin. (**b**) Western blotting of mAb FMC 75 to oligomerized calreticulin. MW: Molecular weight (KDa), lane 1: heat-treated human recombinant CRT (3 days at 42 °C, 57 °C 10 min), lane 2: heat-treated human CRT purified from human placenta (57 °C 10 min) lane 3; heat-treated human recombinant CRT (57 °C 10 min).

**Table 1 antibodies-10-00031-t001:** Overview of peptides used for antibody characterization.

	*N*-Terminal Truncated Peptide	C-Terminal Truncated Peptides	Functionality-Substituted Peptides	Ala-Substituted Peptides
mAb FMC 75				
Template	GDGWTSRWIESKHKSD		TSRWIESK	
	DGWTSRWIESKHKSD	GDGWTSRWIESKHKS	**S**SRWIESK	**A**SRWIESK
	GWTSRWIESKHKSD	GDGWTSRWIESKHK	T**T**RWIESK	T**A**RWIESK
	WTSRWIESKHKSD	GDGWTSRWIESKH	TS**K**WIESK	TS**A**WIESK
	TSRWIESKHKSD	GDGWTSRWIESK	TSR**F**IESK	TSR**A**IESK
	SRWIESKHKSD	GDGWTSRWIES	TSRW**L**ESK	TSRW**A**ESK
	RWIESKHKSD	GDGWTSRWIE	TSRWI**D**SK	TSRWI**A**SK
	WIESKHKSD	GDGWTSRWI	TSRWIE**T**K	TSRWIE**A**K
	IESKHKSD	GDGWTSRW	TSRWIES**R**	TSRWIES**A**
	ESKHKSD	GDGWTSR		
	SKHKSD	GDGWTS		
	KHKSD	GDGWT		
	HKSD	GDGW		
mAb 16				
Template	KDKQDEEQRLKEEEEDKKRK		DEEQRLKEEEED	
	QDEEQRLKEEEEDKK	KQDEEQRLKEEEEDK	**E**EEQRLKEEEED	**A**EEQRLKEEEED
	DEEQRLKEEEEDKK	KQDEEQRLKEEEED	D**D**EQRLKEEEED	D**A**EQRLKEEEED
	EEQRLKEEEEDKK	KQDEEQRLKEEEE	DE**D**QRLKEEEED	DE**A**QRLKEEEED
	EQRLKEEEEDKK	KQDEEQRLKEEE	DEE**N**RLKEEEED	DEE**A**RLKEEEED
	QRLKEEEEDKK	KQDEEQRLKEE	DEEQ**K**LKEEEED	DEEQ**A**LKEEEED
	RLKEEEEDKK	KQDEEQRLKE	DEEQR**I**KEEEED	DEEQR**A**KEEEED
	LKEEEEDKK	KQDEEQRLK	DEEQRL**R**EEEED	DEEQRL**A**EEEED
	KEEEEDKK	KQDEEQRL	DEEQRLK**D**EEED	DEEQRLK**A**EEED
	EEEEDKK	KQDEEQR	DEEQRLKE**D**EED	DEEQRLKE**A**EED
	EEEDKK	KQDEEQ	DEEQRLKEE**D**ED	DEEQRLKEE**A**ED
	EEDKK	KQDEE	DEEQRLKEEE**D**D	DEEQRLKEEE**A**D
	EDKK	KQDE	DEEQRLKEEEE**E**	DEEQRLKEEEE**A**

## Data Availability

All the data obtained in the current study is presented in the manuscript.

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
