# Peer review of "Epitope Mapping of Monoclonal Antibodies to Calreticulin Reveals That Charged Amino Acids Are Essential for Antibody Binding"

_2073-4468, 2021, doi:10.3390/antib10030031_

Round 1
Reviewer 1 Report
The authors use well-established techniques to establish the epitope of two monoclonal antibodies and by the use of modified peptides confirm that positively charged amino-acids are important for the affinity. Nor the experimental approach nor the results present novelty. The data presented are incomplete and it would be interesting to know how the two mabs were isolated. From mice immunized with wild type protein? Or phage display? It would make the paper improved if also other mabs with different affinity and isotype would have been included.
Author Response
Dear reviewer,
thank your for your constructive comments, which have improved the manuscripts. All your comments have been added to the manuscript.
Best regards
Nicole Trier, Gunnar Houen
Reviewer 1:
The authors use well-established techniques to establish the epitope of two monoclonal antibodies and by the use of modified peptides confirm that positively charged amino-acids are important for the affinity. Nor the experimental approach nor the results present novelty. The data presented are incomplete and it would be interesting to know how the two mabs were isolated. From mice immunized with wild type protein? Or phage display? It would make the paper improved if also other mabs with different affinity and isotype would have been included.
Response: In this study we characterized two antibodies commercially available, which have been important in determining the nature and biological functions of calreticulin. Currently no knowledge about the binding sites of these two antibodies has been presented. Information about how the two antibodies were generated has been elaborated in the materials and methods section.
Reviewer 2 Report
This manuscript by Bergmann et al describes the detailed characterization of the linear epitopes of two mAbs that bind to the chaperone Calreticulin (CRT). Using an overlapping peptide library, they determine the approximate location for the linear epitope of each mAb. Then, using truncations, alanine substitutions, and functional substitutions, they determine the amino acids critical for binding of each mAb. While the epitope mapping studies are well done, not much explanation is provided as to why this work is important. Both the introduction and discussion should be improved to make this more clear.
The antibodies under study were not originally identified by this group, but rather were commercially purchased and are listed in the manuscript only as mAb 1 and mAb 2 (though catalog numbers are given). What were the circumstances of these mAbs being previously identified? Why are you studying their epitopes? Do these mAbs have therapeutic potential? Are the identified epitopes potential vaccine targets?
The introduction also mentions that CRT has a characteristic mutation that causes a frameshift in the C-terminus, a change that would certainly affect the epitope of mAb 2. Would this be a concern if mAb 2 were to be used as a therapeutic? The authors also state that CRT can form oligomers, and that that is indeed necessary for function. When CRT oligomerizes, are these epitopes still accessible? Indeed, as an ER resident protein, is CRT even accessible to antibodies?
Finally, the title of the manuscript is incorrect. While the epitope for mAb 1 does indeed rely on two positively charged amino acids (along with other, non-charged amino acids), the epitope for mAb 2 requires several charged amino acids of both positive and negative charges. In the discussion (lines 221-228), it states that “for both epitopes, the charged aas were essential,” which is a more accurate conclusion than the one in the title. However, I would note that this is not exactly surprising for the epitope of mAb 2, since only 2 of its 12 residues are uncharged.
Author Response
Dear reviewer
thank you for your comments to the manuscript. We have addressed all your comments and they have significantly improved the manuscript.
Best regards
Nicole Trier, Gunnar Houen
Reviewer 2:
This manuscript by Bergmann et al describes the detailed characterization of the linear epitopes of two mAbs that bind to the chaperone Calreticulin (CRT). Using an overlapping peptide library, they determine the approximate location for the linear epitope of each mAb. Then, using truncations, alanine substitutions, and functional substitutions, they determine the amino acids critical for binding of each mAb. While the epitope mapping studies are well done, not much explanation is provided as to why this work is important. Both the introduction and discussion should be improved to make this more clear.
Response: We agree that the point of the manuscript could be described further. The importance of the antibodies has been elaborated in the introduction and the discussion.
The antibodies under study were not originally identified by this group, but rather were commercially purchased and are listed in the manuscript only as mAb 1 and mAb 2 (though catalog numbers are given). What were the circumstances of these mAbs being previously identified? Why are you studying their epitopes? Do these mAbs have therapeutic potential? Are the identified epitopes potential vaccine targets?
Response: Antibody studies have been very central in relation to the nature and biological functions of CRT. Currently no knowledge about the binding sites of these two antibodies has been presented. Moreover, knowledge about the interaction between CRT and mAb 16 may aid in elucidating the structure of CRT, as only parts of the CRT structure has been crystalized. Moreover, the mAb FMC 75 may have therapeutic potential, as it recognized CRT in the oligomeric structure, which is essential for some CRT functions. This has been elaborated. Based on these findings, the mAb 16 and/or its epitope not have a therapeutic/vaccine potential.
The introduction also mentions that CRT has a characteristic mutation that causes a frameshift in the C-terminus, a change that would certainly affect the epitope of mAb 2. Would this be a concern if mAb 2 were to be used as a therapeutic? The authors also state that CRT can form oligomers, and that that is indeed necessary for function. When CRT oligomerizes, are these epitopes still accessible? Indeed, as an ER resident protein, is CRT even accessible to antibodies?
Response: The mAb FMC 75 recognizes CRT oligomers, confirming that the epitope still is accessible. CRT is accessible to the antibodies when exposed on the surface of cells. The second mAb (mAb 16) does not recognize frameshifted CRT.
Finally, the title of the manuscript is incorrect. While the epitope for mAb 1 does indeed rely on two positively charged amino acids (along with other, non-charged amino acids), the epitope for mAb 2 requires several charged amino acids of both positive and negative charges. In the discussion (lines 221-228), it states that “for both epitopes, the charged aas were essential,” which is a more accurate conclusion than the one in the title. However, I would note that this is not exactly surprising for the epitope of mAb 2, since only 2 of its 12 residues are uncharged.
Response: A more appropriate title has been selected.
Reviewer 3 Report
The authors characterize two peptide epitopes for two commercially available mouse calreticulin binding antibodies FMC 75 and 612137. Using an overlapping peptide library the authors identify the sequences both antibodies bind to and then further characterize binding with alanine scanning mutagenesis of those sequences. Calreticulin has many cellular functions and has been implicated in certain cancers. Use of the antibodies FMC 75 and 612137 could help inform calreticulin research. The paper could be significantly improved however if the authors were to relate their results back to calreticulin research. Have these commercially available antibodies been used in any published research? Would identification of the antibody epitopes presented here add anything to the interpretation of that research? Or alternatively could the authors use either of these antibodies in calreticulin research of their own?
Some minor points. Referring to the antibodies as mAb1 and mAb2 is confusing and annoying. FMC 75 and 612137 are not great names but at least they are unique. Imagine if every paper referred to their antibodies as 1 and 2. Also antibodies bind their epitope. Reactivity refers to how the binding is detected.
Author Response
Dear reviewer
thank you for your comments to the manuscript. We have addressed your comments and they have significantly improved the manuscript.
Best regards
Nicole Trier, Gunnar Houen
Reviewer 3:
The authors characterize two peptide epitopes for two commercially available mouse calreticulin binding antibodies FMC 75 and 612137. Using an overlapping peptide library the authors identify the sequences both antibodies bind to and then further characterize binding with alanine scanning mutagenesis of those sequences. Calreticulin has many cellular functions and has been implicated in certain cancers. Use of the antibodies FMC 75 and 612137 could help inform calreticulin research. The paper could be significantly improved however if the authors were to relate their results back to calreticulin research. Have these commercially available antibodies been used in any published research? Would identification of the antibody epitopes presented here add anything to the interpretation of that research? Or alternatively could the authors use either of these antibodies in calreticulin research of their own?
Some minor points. Referring to the antibodies as mAb1 and mAb2 is confusing and annoying. FMC 75 and 612137 are not great names but at least they are unique. Imagine if every paper referred to their antibodies as 1 and 2. Also antibodies bind their epitope. Reactivity refers to how the binding is detected.
Response: Amended as requested, the names of the mAbs have been changed to the clone numbers. The antibodies have been used in several studies, where the functions of CRT have been investigated. This has been added to the manuscript and related to therapeutic use.
Round 2
Reviewer 1 Report
By making the study more informative about possible targets in calreticulin the paper gains in interest